# Data-Efficient and Robust Coreset Selection via Sparse Adversarial Perturbations

**Tushar Shinde, Manasa Madabhushi**
MIDAS (Multimedia Intelligence, Data Analysis and compreSsion) Lab
Indian Institute of Technology Madras, Zanzibar, Tanzania
shinde@iitmz.ac.in

## Abstract

Efficient training of deep neural networks under data constraints relies on selecting informative subsets, or coresets, that preserve model performance. Traditional methods in libraries like DeepCore employ heuristics such as uncertainty sampling or gradient diversity but often neglect adversarial vulnerabilities, leading to suboptimal robustness against distribution shifts, corruptions, or manipulations in unreliable data scenarios. To address this, we introduce a unified Adversarial Sensitivity Scoring framework comprising three novel ranking techniques: Inverse Sensitivity and Entropy Fusion (ISEF), Fast Gradient Sign Method with Composite Scoring (FGSM-CS), and Perturbation Sensitivity Scoring (PSS), that harness sparse adversarial perturbations to prioritize samples near decision boundaries. By applying single-step Sparse FGSM attacks, our methods expose sample sensitivities with minimal computational overhead. Evaluated on CIFAR-10 with ResNet-18, our approaches consistently outperform the adversarial baseline DeepFool by up to 15.1% in extremely sparse data regimes (e.g., 1% for $PSS_{bottom}$) and 15.1% in low data regimes (e.g., 10% for $FGSM\text{-}CS_{bottom}$), while achieving comparable results to top DeepCore methods like Random and Forgetting in moderate data regimes. Notably, bottom variants excel in sparse settings by retaining perturbation-resilient samples, with top variants surpassing them after 20–30% boundaries. These gains, realized via efficient single-step gradients, position our framework as a scalable, deployable bridge between coreset selection and adversarial robustness, advancing data-efficient learning.

## 1 Introduction and Related Work

As deep learning models grow in size and complexity, improving data efficiency and robustness remains a fundamental challenge. One promising direction to address these challenges is *coreset selection*, where a small, informative subset of training data is identified to reduce computational costs while maintaining or even improving model performance Shinde [2025]. Traditional approaches, such as DeepCore Guo et al. [2022], utilize heuristics and optimization techniques to select representative samples, demonstrating success in data subset selection, continual learning Castro et al. [2018], Shinde and Sharma, and active learning Gal et al. [2017]. These methods aim to emphasize sample diversity and informativeness, thereby capturing the essential characteristics of the dataset. However, existing coreset selection techniques often overlook adversarial vulnerabilities, leaving models susceptible to unreliable data scenarios such as distribution shifts, corruptions, or strategic manipulations, as seen in real-world applications like autonomous systems or social platforms Hardt et al. [2016], Hassan and Shinde.

Recent advancements in adversarial machine learning have highlighted the vulnerability of deep networks to small, sparse perturbations. Notably, the One Pixel Attack Su et al. [2019] demonstrated that modifying even a single pixel can lead to misclassification, emphasizing the sensitivity of deep

models near decision boundaries. This underscores the need to prioritize these regions, which are critical for both learning efficiency and robustness. Our work extends this by using sparse adversarial perturbations as a signal for coreset construction, explicitly targeting decision boundary samples to enhance model reliability under adversarial conditions.

In addition, a few recent studies have explored the intersection of coreset selection and adversarial robustness Mirzasoleiman et al. [2020]. However, these studies either focus on improving robustness without considering data efficiency or improve data efficiency without directly addressing adversarial robustness. Our work bridges this gap by presenting a unified Adversarial Sensitivity Scoring framework that leverages sparse adversarial perturbations to simultaneously enhance efficiency and robustness, offering a principled approach to tackle coreset data challenges.

In this work, we introduce a novel strategy that leverages *sparse adversarial perturbations* to identify and prioritize training samples near the decision boundary, which are essential for both learning informativeness and robustness. Specifically, we employ the Sparse Fast Gradient Sign Method (Sparse FGSM) Goodfellow et al. [2014], a variant of the Fast Gradient Sign Method (FGSM), which generates adversarial examples by perturbing only a small subset of the input features (e.g., pixels). This method produces *minimally sparse* yet highly informative perturbations that highlight samples that are vulnerable to model changes and adversarial attacks, making them ideal candidates for robust and data-efficient training in scenarios with corrupted or manipulated data Hendrycks and Dietterich [2019]. The effectiveness of this approach is grounded in recent work on *adversarial sensitivity*, which shows that deep networks are particularly sensitive to small perturbations near the decision boundary Madry et al. [2017]. By focusing on these perturbations, we can identify samples that are both highly informative and critical for improving adversarial robustness Wong et al. [2020].

In this direction, we propose three ranking schemes under our Adversarial Sensitivity Scoring framework, systematically categorized by their sensitivity metrics, based on the behavior of samples under Sparse FGSM attacks. We evaluate the performance of these schemes by selecting both the *top-ranked* samples (those most vulnerable or near the decision boundary) and the *bottom-ranked* samples (those least vulnerable, far from the boundary). We conduct extensive experiments on CIFAR-10, varying the selection ratio from 0.1% to 90% of the data to assess the effectiveness of our approach across different data budgets. Our results demonstrate the advantage of focusing on the *top* samples, which result in substantial improvements in classification accuracy and adversarial robustness, especially in low-data regimes.

We benchmark our method against existing coreset selection methods implemented in the DeepCore library Guo et al. [2022], including DeepFool-based selection Ducoffe and Precioso [2018]. In a range of experimental settings, our Sparse Adversarial Ranking consistently outperforms the DeepFool-based method and achieves performance comparable to the overall DeepCore library. In particular, our approach demonstrates superior adversarial robustness while maintaining high computational efficiency. This paper presents a novel framework for coreset selection that utilizes sparse adversarial perturbations to identify vulnerable and informative samples, offering a scalable solution, with potential applications in safety-critical domains.

The rest of the paper is organized as follows: Section 2 introduces the proposed architecture and modeling pipeline, Section 3 outlines the dataset, preprocessing, and training setup, Section 4 reports the experimental results, and Section 5 concludes with future directions.

## 2 Method

To construct an informative coreset guided by adversarial sensitivity, we employ the Sparse Fast Gradient Sign Method (Sparse FGSM) to perturb training images at varying levels of sparsity $k$. From each class, we extract a subset of samples ranked using three distinct yet complementary techniques within our unified Adversarial Sensitivity Scoring framework, systematically categorized by their sensitivity metrics (entropy-based, composite vulnerability, and perturbation-shift), designed to evaluate each sample's adversarial vulnerability and overall contribution to model robustness, thereby addressing unreliable data challenges such as distribution shifts and strategic manipulations by prioritizing samples that expose model weaknesses.

We begin by training a Deep Neural Network on the CIFAR-10 dataset. To evaluate its robustness, we apply Sparse FGSM, an adaptation of the classic Fast Gradient Sign Method (FGSM), which generates

adversarial examples by perturbing only a subset of pixels with the highest gradient magnitudes, as inspired by sparse attack literature Dinh et al. [2020]. The perturbed images are fed into the trained model to assess whether classification changes occur, indicating sensitivity to sparse adversarial noise that mimics real-world corruptions. For each perturbation level $k$, we compute and store the resulting class-wise probability distributions in a structured CSV format for subsequent analysis.

To investigate data efficiency and sample informativeness, we propose a novel ranking framework that leverages these output distributions to guide coreset selection, building on prior coreset methods Sener and Savarese [2017] while incorporating adversarial cues to enhance robustness. Unlike traditional methods, our framework targets samples near decision boundaries to improve generalization, using a pre-trained model to initialize selection, which can be adapted via transfer learning to mitigate the need for full-dataset training. We then retrain the model using only the selected coreset and compare its performance to a baseline trained on the full dataset to determine gains in generalization and robustness. This pipeline provides a comprehensive framework for evaluating both adversarial resilience and the role of informed sample selection in deep learning, offering reliable ML solutions under unreliable data constraints.

## 2.1 Sparse FGSM Technique

To investigate model robustness against sparse adversarial perturbations that simulate localized data corruptions in unreliable environments Modas et al. [2019], we adopt a variant of the Fast Gradient Sign Method (FGSM) Goodfellow et al. [2014] known as **Sparse FGSM**, which aligns with recent advances in efficient attack generation Tramèr et al. [2017]. Unlike the conventional FGSM, which perturbs all input pixels, Sparse FGSM restricts modifications to a fixed number $k$ of pixels with the largest gradient magnitudes, thereby simulating more realistic and less perceptible adversarial scenarios, such as those encountered in distribution-shifted or manipulated data Hendrycks et al. [2021]. Formally, given an input $x \in \mathbb{R}^n$, a true label $y$, and a loss function $\mathcal{L}(f(x), y)$, the standard FGSM constructs an adversarial example as: $x^{\text{adv}} = x + \epsilon \cdot \text{sign}\left(\nabla_x \mathcal{L}(f(x), y)\right)$, where $\epsilon$ is the perturbation budget and $f(\cdot)$ denotes the model's predictive function. In Sparse FGSM, the perturbation is instead applied only to a subset $\mathcal{K} \subset \{1, 2, \ldots, n\}$, corresponding to the indices of the top-$k$ largest absolute gradient magnitudes:

$$x_i^{\text{adv}} = \begin{cases} x_i + \epsilon \cdot \text{sign}(\nabla_{x_i} \mathcal{L}(f(x), y)), & \text{if } i \in \mathcal{K}, \\ x_i, & \text{otherwise.} \end{cases} \tag{1}$$

This sparsity constraint introduces a more localized perturbation strategy, preserving most of the input while targeting only the most influential dimensions, which intuitively reveals samples' vulnerability to minimal changes, enhancing coreset selection by prioritizing samples that strengthen decision boundaries against strategic manipulations or corruptions Hardt et al. [2016]. In our experiments, we evaluate the model's behavior under varying levels of sparsity by selecting $k \in \{1, 4, 16, 64\}$. These values range from extremely sparse ($k = 1$) to moderately sparse ($k = 64$), enabling a fine-grained analysis of the degradation in model performance under localized adversarial threats and providing insights into why robustness degrades more sharply for certain classes..

## 2.2 Coreset Selection Techniques

Training modern deep neural networks on large datasets is computationally intensive and often involves redundant or less informative samples. Coreset selection seeks to identify a representative subset of the training data that maintains model performance while reducing training cost Feldman [2019]. However, in adversarial settings, traditional coreset methods may overlook the varying robustness and uncertainty of individual samples, which are critical for enhancing generalization and adversarial resilience.

To address this, we propose coreset selection strategies under our Adversarial Sensitivity Scoring framework, systematically categorized into entropy-based (ISEF), composite vulnerability (FGSM-CS), and perturbation-shift (PSS) metrics, leveraging each sample's sensitivity to adversarial perturbations alongside predictive uncertainty, enhancing robustness by selecting samples that fortify decision boundaries against distribution shifts and manipulations Madry et al. [2017]. Our approaches integrate metrics such as misclassification frequency, confidence drop, entropy, and perturbation-induced output variation. Specifically, we develop three complementary ranking techniques: Inverse Sensitivity and

Entropy Fusion (ISEF), which balances perturbation sensitivity and entropy; FGSM-Based Composite Scoring (FGSM-CS), which combines adversarial vulnerability, uncertainty, and robustness into a weighted composite score; and Perturbation Sensitivity Score (PSS), which quantifies the cumulative effect of localized pixel perturbations on model outputs, with the key motivation that these metrics identify samples near adversarial frontiers, improving robustness and efficiency.

### 2.2.1 Inverse Sensitivity and Entropy Fusion (ISEF)

We propose a coreset selection strategy that ranks training samples using a composite score that combines adversarial sensitivity and predictive uncertainty, motivated by the need to prioritize samples near decision boundaries for better handling of distribution shifts. Each sample $\mathbf{x}$ is assigned a composite score $S(\mathbf{x})$ defined as:

$$S(\mathbf{x}) = \alpha \cdot \tilde{S}_{\text{inv}_k}(\mathbf{x}) + (1 - \alpha) \cdot \tilde{S}_{\text{ent}}(\mathbf{x}), \tag{2}$$

where $\alpha \in [0, 1]$ is a weighting parameter (set to $\alpha = 0.5$ in our experiments), and $\tilde{S}_{\text{inv}_k}$, $\tilde{S}_{\text{ent}}$ are the min-max normalized versions of the following metrics. The inverse sensitivity score $S_{\text{inv}_k}$ is defined as 1 if misclassified on the clean input, $\frac{1}{k_{\min}}$ if misclassified at some $k \in \{1, 4, 16, 64\}$ where $k_{\min}$ is the smallest perturbation level that causes misclassification, or 0 if correctly classified for all $k$, intuitively rewarding early misclassifications as indicators of inherent vulnerability. The entropy score $S_{\text{ent}}$ is the Shannon entropy of the model's softmax output on the clean input, computed as $-\sum_{i=1}^{C} p_i \log_{10}(p_i)$, where $p_i$ is the predicted probability for class $i$, and $C$ is the number of classes, capturing uncertainty that complements sensitivity for robust coreset selection.

### 2.2.2 FGSM-Based Composite Scoring (FGSM-CS)

To comprehensively assess model robustness under adversarial perturbations, we propose a composite scoring mechanism that integrates multiple vulnerability indicators into a single metric. This approach acknowledges that robustness cannot be adequately characterized by accuracy alone; it must also reflect the model's sensitivity to perturbations, predictive uncertainty, and consistency in maintaining correct classifications under adversarial conditions.

The composite score $S$ is defined as a weighted combination of three components: adversarial sensitivity ($S_{\text{adv}}$), uncertainty ($S_{\text{unc}}$), and binary robustness ($S_{\text{rob}}$) and computed as:

$$S = w_{\text{adv}} \cdot S_{\text{adv}} + w_{\text{unc}} \cdot S_{\text{unc}} + w_{\text{rob}} \cdot S_{\text{rob}}, \tag{3}$$

where the weights are empirically set as $w_{\text{adv}} = 0.5$, $w_{\text{unc}} = 0.3$, and $w_{\text{rob}} = 0.2$, chosen to emphasize sensitivity while balancing other factors for practical deployment. To compute $S_{\text{adv}}$, we apply Sparse Fast Gradient Sign Method (FGSM) attacks with varying pixel-level perturbation sizes $k \in \{1, 4, 16, 64\}$. Let $p^{(0)}$ denote the softmax output of the model on the clean input, and $p^{(k)}$ represent the output on the adversarially perturbed input with perturbation level $k$. The adversarial sensitivity is then defined as:

$$S_{\text{adv}} = \sum_{k \in \{1,4,16,64\}} \alpha_k \cdot \left\| p^{(0)} - p^{(k)} \right\|_1, \tag{4}$$

where $\alpha_k$ is a decay weight (e.g., $\alpha_k = \frac{1}{k}$) that emphasizes the model's vulnerability to smaller perturbations. The uncertainty component $S_{\text{unc}}$ captures the maximum softmax entropy across the clean and adversarial variants of the input:

$$S_{\text{unc}} = \max_{k \in \{0,1,4,16,64\}} \left( -\sum_{i=1}^{C} p_i^{(k)} \log p_i^{(k)} \right), \tag{5}$$

where $p_i^{(k)}$ is the predicted probability of class $i$ under perturbation level $k$, and $C$ is the total number of classes. Finally, the binary robustness indicator $S_{\text{rob}}$ is defined as:

$$S_{\text{rob}} = \begin{cases} 1, & \text{if } \exists k \in \{1, 4, 16, 64\} \text{ such that } \hat{y}^{(k)} \neq y_{\text{true}}, \\ 0, & \text{otherwise,} \end{cases} \tag{6}$$

where $\hat{y}^{(k)}$ is the predicted class label at perturbation level $k$, and $y_{\text{true}}$ is the ground-truth label. A higher composite score $S$ indicates greater model fragility, reflecting increased susceptibility to adversarial perturbations, elevated predictive uncertainty, and reduced robustness in classification consistency.

### 2.2.3 Perturbation Sensitivity Score (PSS)

In adversarial and robustness-aware training, it is crucial to identify inputs for which a model's predictions are highly sensitive to small, localized changes. These inputs often lie near the decision boundary and can expose weaknesses in the model's generalization capability under imperfect data Biggio et al. [2013]. To quantify this sensitivity, we introduce the Perturbation Sensitivity Score (PSS), a metric designed to measure how much a model's output distribution shifts under sparse, localized pixel perturbations.

To evaluate the sensitivity of images to localized pixel perturbations, let $\mathcal{K} = \{1, 4, 16, 64\}$ denote the set of pixel perturbation sizes. For each image $x_i$, and for each $k \in \mathcal{K}$, we apply a $k$-pixel perturbation and compute the softmax output $p_k(x_i) \in \mathbb{R}^{10}$ from the classifier. Let $p_0(x_i)$ be the original (unperturbed) softmax prediction. We define the *Perturbation Sensitivity Score (PSS)* for image $x_i$ as:

$$\text{PSS}(x_i) = \sum_{k \in \mathcal{K}} \frac{1}{\log_2(k) + 1} \cdot \|p_k(x_i) - p_0(x_i)\|_2 \tag{7}$$

The score captures the cumulative change in class probabilities due to small, localized perturbations. The $\log_2(k + 1)$ term downweights the influence of large $k$, emphasizing sensitivity to minimal perturbations. A higher PSS indicates that the image is more fragile under small changes and thus likely resides near the decision boundary. These images are prioritized in the coreset due to their potential to improve generalization during training.

## 3 Experimental Setup

The experiment was conducted on the Kaggle platform using an NVIDIA Tesla P100 GPU. We evaluate model performance through adversarial robustness under various perturbation conditions and coreset selection strategies.

### 3.1 Dataset

The CIFAR-10 dataset consists of 60,000 color images uniformly distributed across 10 distinct object categories: airplane, automobile, bird, cat, deer, dog, frog, horse, ship, and truck. Each image has a size of 32×32 pixels with 3 color channels (RGB). The dataset is split into 50,000 training images and 10,000 test images, ensuring that the performance can be robustly evaluated.

### 3.2 Model Training Setup

**Baseline Model.** We begin by training a baseline ResNet-18 model on the CIFAR-10 training dataset. All experiments adopt the ResNet-18 architecture, consistent with prior coreset work Guo et al. [2022], ensuring a fair baseline comparison. While other architectures may yield higher baseline performance, our analysis focuses on generalizable insights, validated through extensive experiments across varying data fractions and perturbation levels.

**Model Training.** For the CIFAR-10 experiments, we trained a ResNet-18 architecture using stochastic gradient descent (SGD) with a batch size of 128, an initial learning rate of 0.1, momentum of 0.9, and a weight decay of $5 \times 10^{-4}$. The learning rate was scheduled with cosine annealing over 200 epochs. Standard data augmentation techniques were applied, including random cropping with padding = 4 and horizontal flips, along with CIFAR-10 normalization (mean = (0.4914, 0.4822, 0.4465), std = (0.2023, 0.1994, 0.2010)).

**Adversarial Robustness Setup.** To assess adversarial robustness, we generate Sparse FGSM perturbations at varying sparsity levels $k \in \{0, 1, 4, 16, 64\}$. Increasing values of $k$ correspond to progressively higher sparsity, simulating more localized adversarial attacks on the images. For each sparsity level, we compute the predicted class probabilities of all training samples. These probabilities, along with the corresponding predicted labels and confidence scores, are stored in CSV files for subsequent analysis.

**Model Training on the Coreset.** We further evaluate performance by training models on selected subsets (coresets) of the CIFAR-10 dataset. The subset sizes vary across fractions of the full dataset:

0.1%, 0.5%, 1%, 5%, 10%, 20%, 30%, 40%, 50%, 60%, and 90%. Based on the ranking scores stored in the CSV files, two types of coresets are constructed for each fraction: (1) *Top-ranked subsets*, consisting of samples identified as highly vulnerable or uncertain, and (2) *Bottom-ranked subsets*, consisting of samples deemed robust or confident. To ensure class balance and avoid selection bias, the number of samples chosen from each class is capped proportionally.

### 3.3 Hyperparameter Settings

For the **ISEF** method, we set the balancing factor to $\alpha = 0.5$. For **FGSM-CS**, the weighting parameters were chosen as $w_{adv} = 0.5$, $w_{unc} = 0.3$, and $w_{rob} = 0.2$. To evaluate adversarial robustness under varying levels of perturbation, we applied Sparse FGSM with sparsity levels $k \in \{1, 4, 16, 64\}$, corresponding to progressively stronger pixel perturbations.

We use classification accuracy as the primary evaluation metric to assess model performance. The accuracy is evaluated across different perturbation levels to quantify adversarial robustness.

## 4 Results and Analysis

We assess performance across varying data fractions, focusing on both the top (samples most vulnerable to adversarial attacks) and bottom (least sensitive) subsets, providing a unified view of how our Adversarial Sensitivity Scoring framework outperforms by prioritizing critical samples.

### 4.1 Ablation Study

We conduct an extensive ablation study to evaluate the effectiveness of our proposed ranking techniques: ISEF, FGSM-CS, and PSS, on CIFAR-10, using varying fractions of the training data from 0.1% to 90%. Table 1 summarizes the classification accuracy achieved by each method across different data fractions, illustrating how adversarially-informed selection enhances model performance, particularly in low-data regimes, by prioritizing samples that mitigate biases, label noise, and vulnerabilities. The ablation results are integrated into the main comparison table to avoid redundancy, focusing on our methods' performance relative to DeepCore baselines.

#### 4.1.1 Comparative Analysis: Different Data Regimes

**Extremely Sparse Data Regimes (0.1% - 1%).** At extremely sparse data fractions, $PSS_{bottom}$ consistently outperforms other variants, e.g., achieving 33.3% at 0.5% and 40.6% at 1%. This demonstrates its ability to identify near-decision-boundary samples that are sensitive to minimal perturbations, emphasizing influential pixels that reveal intrinsic model fragility. $ISEF_{bottom}$ also excels (19.9% at 0.1%, 38.4% at 1%), as its entropy-rich samples provide diverse, stable information, aligning with data valuation strategies that prioritize robust coverage.

**Low-to-Moderate Data Regimes (5% - 20%).** In the low-to-moderate data regimes, $FGSM-CS_{bottom}$ dominates, e.g., 75.9% at 10%, followed by $ISEF_{bottom}$ (75.0%). This underscores the effectiveness of selecting non adversarial samples: the single-step gradient perturbation efficiently identifies examples that offer strong learning signals without requiring expensive iterative attacks Ducoffe and Precioso [2018], capturing multifaceted vulnerability through a composite score that blends sensitivity, uncertainty, and robustness, preventing overfitting to noisy data.

**Moderate-to-High Data Regimes (30% - 90%).** For moderate-to-high data fractions, $ISEF_{top}$ frequently matches or surpasses other methods (e.g., 93.3% at 40%, 95.4% at 90%). Across these higher fractions, all methods converge above 94.7% accuracy, indicating that the influence of individual sample selection diminishes with sufficient data.

#### 4.1.2 Comparative Analysis: Top vs. Bottom Variants

Figure 1 and Table 1 highlight the nuanced trade-offs between top- and bottom-ranked variants across varying dataset fractions. At very small fractions (e.g., 0.1%–1%), bottom variants such as $ISEF_{bottom}$ and $PSS_{bottom}$ clearly dominate, with $PSS_{bottom}$ reaching 33.3% and 40.6% accuracy at 0.5% and 1%. This trend suggests that low-entropy or "easy" samples provide a more stable foundation for bootstrapping learning in severely data-limited regimes, as they reduce variance and

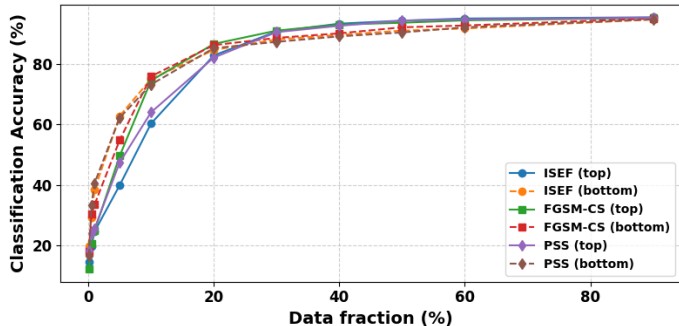

Figure 1: Performance of top vs. bottom variants across data fractions on CIFAR-10 for ISEF, FGSM-CS, and PSS, highlighting the transition where top variants overtake bottom variants around 20–30%, reflecting the shift from stable, entropy-rich samples to perturbation-sensitive samples.

Table 1: Classification accuracy (in %) of novel ranking techniques compared with DeepCore library methods Guo et al. [2022] on CIFAR-10. Best, second, and third values in each column are highlighted using green shades.

| Method | 0.1% | 0.5% | 1% | 5% | 10% | 20% | 30% | 40% | 50% | 60% | 90% |
|---|---|---|---|---|---|---|---|---|---|---|---|
| Random Guo et al. [2022] | 21.0 | 30.8 | 36.7 | 64.5 | 75.7 | 87.1 | 90.2 | 92.1 | 93.3 | 94.0 | 95.2 |
| CD Agarwal et al. [2020] | 15.8 | 20.5 | 23.6 | 38.1 | 58.8 | 81.3 | 90.8 | 93.3 | 94.3 | 94.6 | 95.4 |
| Herding Welling [2009] | 20.2 | 27.3 | 34.8 | 51.0 | 63.5 | 74.1 | 80.1 | 85.2 | 88.0 | 89.8 | 94.6 |
| k-Center Greedy Sener and Savarese [2017] | 18.5 | 26.8 | 31.1 | 51.4 | 75.8 | 87.0 | 90.9 | 92.8 | 93.9 | 94.1 | 95.4 |
| Least Confidence Coleman et al. [2019] | 14.2 | 17.2 | 19.8 | 36.2 | 57.6 | 81.9 | 90.3 | 93.1 | 94.5 | 94.7 | 95.5 |
| Entropy Coleman et al. [2019] | 14.6 | 17.5 | 21.1 | 35.3 | 57.6 | 81.9 | 89.8 | 93.2 | 94.4 | 95.0 | 95.4 |
| Margin Coleman et al. [2019] | 17.2 | 21.7 | 28.2 | 43.4 | 59.9 | 81.7 | 90.9 | 93.0 | 94.3 | 94.8 | 95.5 |
| Forgetting Toneva et al. [2018] | 21.4 | 29.8 | 35.2 | 52.1 | 67.0 | 86.6 | 91.7 | 93.5 | 94.1 | 94.6 | 95.3 |
| GraNd Paul et al. [2021] | 17.7 | 24.0 | 26.7 | 39.8 | 52.7 | 78.2 | 91.2 | 93.7 | 94.6 | 95.0 | 95.5 |
| Cal Margatina et al. [2021] | 22.7 | 33.1 | 37.8 | 60.0 | 71.8 | 80.9 | 86.0 | 87.5 | 89.4 | 91.6 | 94.7 |
| Craig Mirzasoleiman et al. [2020] | 22.5 | 27.0 | 31.7 | 45.2 | 60.2 | 79.6 | 88.4 | 90.8 | 93.3 | 94.2 | 95.5 |
| GradMatch Killamsetty et al. [2021a] | 17.4 | 25.6 | 30.8 | 47.2 | 61.5 | 79.9 | 87.4 | 90.4 | 92.9 | 93.2 | 93.7 |
| Glister Killamsetty et al. [2021b] | 19.5 | 27.5 | 32.9 | 50.7 | 66.3 | 84.8 | 90.9 | 93.0 | 94.0 | 94.8 | 95.6 |
| FL Iyer et al. [2021] | 22.3 | 31.6 | 38.9 | 60.8 | 74.7 | 85.6 | 91.4 | 93.2 | 93.9 | 94.5 | 95.5 |
| DeepFool Ducoffe and Precioso [2018] | 17.6 | 22.4 | 27.6 | 42.6 | 60.8 | 83.0 | 90.0 | 93.1 | 94.1 | 94.8 | 95.5 |
| ISEF$_{top}$ | 14.5 | 19.8 | 24.6 | 39.9 | 60.3 | 82.8 | 90.7 | 93.3 | 94.3 | 95.0 | 95.4 |
| ISEF$_{bottom}$ | 19.9 | 29.4 | 38.4 | 62.8 | 75.0 | 84.6 | 88.1 | 89.6 | 91.0 | 91.7 | 94.7 |
| FGSM-CS$_{top}$ | 12.2 | 20.5 | 24.9 | 49.6 | 74.5 | 86.6 | 91.0 | 93.0 | 93.6 | 94.4 | 95.2 |
| FGSM-CS$_{bottom}$ | 17.9 | 30.3 | 33.5 | 54.8 | 75.9 | 86.2 | 88.6 | 90.1 | 92.1 | 92.7 | 95.0 |
| PSS$_{top}$ | 18.3 | 24.0 | 25.6 | 47.4 | 64.0 | 82.1 | 90.5 | 92.6 | 94.3 | 94.7 | 95.3 |
| PSS$_{bottom}$ | 16.6 | 33.3 | 40.6 | 62.2 | 73.2 | 85.2 | 87.3 | 89.1 | 90.4 | 92.1 | 94.7 |

facilitate class separability early in training. As the dataset fraction grows, however, the advantage shifts toward the top-ranked variants. ISEF$_{top}$ surpasses ISEF$_{bottom}$ beyond 20%, ultimately peaking at **95.0**% at 60% and **95.4**% at 90%, emphasizing the importance of "hard" decision-boundary samples for refining classification performance once a stable backbone is established. Similarly, FGSM-CS$_{top}$ demonstrates consistent superiority in moderate regimes (20%–30%), attaining 86.6% at 20% and 91.0% at 30%, which highlights the efficiency of targeting adversarially vulnerable points that uncertainty-only methods often overlook.

Finally, in high data regimes (50%–90%), PSS$_{top}$ outperforms PSS$_{bottom}$, reaching 95.3% at 90%. This inversion underscores that perturbation-informed scoring not only accelerates learning in moderate regimes. Collectively, these results indicate a two-phase dynamic: bottom variants help bootstrap robust representations in scarce-data settings, while top variants progressively dominate as sample diversity increases and the decision boundary requires sharper refinement.

## 4.2 Comparison with Existing Work

To evaluate the efficacy of our proposed novel ranking schemes: ISEF, FGSM-CS, and PSS, which leverage sparse adversarial perturbations for coreset selection, we adopt the experimental setup from the baseline DeepCore library Guo et al. [2022], employing a ResNet-18 architecture trained on CIFAR-10. Table 1 reports the test accuracies across varying data fractions, with the last six

rows highlighting our methods. These results demonstrate that our adversarial-informed approaches excel in identifying informative samples, particularly in low-data regimes, by prioritizing those that expose model vulnerabilities and mitigate biases or corruptions. In contrast to traditional methods in DeepCore, which often rely on uncertainty sampling (e.g., Entropy, Margin) or gradient-based diversity (e.g., GraNd, GradMatch), our techniques integrate perturbation sensitivity to favor samples that are "hard" under small corruptions. This not only enhances generalization but also reduces computational demands, as our methods avoid iterative optimizations per sample.

### 4.2.1 Comparison under Different Data Regimes

We dissect the performance across data regimes to provide granular insights, revealing how our methods' strengths manifest as data availability changes.

**Extremely Sparse Data Regimes (0.1%–1%).** In highly constrained settings, where only a minuscule fraction of data is selected, our methods shine by focusing on perturbation-sensitive samples that capture core model uncertainties. For instance, at 0.1%, Cal Margatina et al. [2021] leads with 22.7%, followed closely by Craig (22.5%) and FL (22.3%), but $ISEF_{bottom}$ achieves a competitive 19.9% while outperforming methods like Entropy (14.6%) and Margin (17.2%). By 0.5% and 1%, $PSS_{bottom}$ surges to the top with 33.3% and 40.6%, surpassing Cal (33.1%) and FL (38.9%). This superiority stems from PSS's emphasis on samples with maximal perturbation-induced score shifts, which intuitively selects points critical for rapid convergence in data-scarce scenarios, addressing challenges like label noise or strategic manipulations.

**Low-to-Moderate Data Regimes (5%–20%).** As data fractions increase, the benefits of adversarial ranking become more pronounced, with our methods often ranking among the top performers. At 5%, Random surprisingly leads with 64.5%, but $ISEF_{bottom}$ (62.8%) and $PSS_{bottom}$ (62.2%) closely follow, outperforming sophisticated methods like GraNd (39.8%) and Cal (60.0%). By 10%, $FGSM\text{-}CS_{bottom}$ achieves the highest accuracy of 75.9%, edging out k-Center Greedy (75.8%) and Random (75.7%), while $ISEF_{bottom}$ (75.0%) and $FGSM\text{-}CS_{top}$ (74.5%) remain competitive. At 20%, Random again tops at 87.1%, but our methods like $FGSM\text{-}CS_{top}$ and Forgetting tie at 86.6%, demonstrating robustness. The FGSM-CS's composite scoring integrates gradient norms with uncertainty, selecting diverse subsets that counteract overfitting to noisy or biased data, outperforming diversity-based approaches like Herding (74.1%) by focusing on adversarial frontiers.

**Moderate-to-High Data Regimes (30%–90%).** In regimes with ample data, performance saturates across methods, as larger coresets inherently capture sufficient representation. Forgetting leads at 30% with 91.7%, but FL (91.4%) and GraNd (91.2%) are close, with $FGSM\text{-}CS_{top}$ at 91.0%. From 40% onward, GraNd dominates (e.g., 93.7% at 40%, 94.6% at 50%), yet our methods like $ISEF_{top}$ (95.0% at 60%) match or exceed Entropy (95.0% at 60%). At 90%, Glister achieves 95.6%, but several of our variants reach 95.0%–95.4%. This convergence suggests that adversarial cues provide limited advantage with abundant data, but our methods' efficiency persists, offering comparable accuracy with lower selection costs.

### 4.3 Comparison with DeepFool Ducoffe and Precioso [2018]

DeepFool, an iterative adversarial method that computes minimal perturbations to fool the model, serves as a natural baseline for our sparse perturbation-based approaches. Table 1 reveals consistent advantages for our methods, especially in sparse regimes. At 0.1%, $ISEF_{bottom}$ (19.9%) outperforms DeepFool (17.6%); by 0.5% and 1%, $PSS_{bottom}$ (33.3%, 40.6%) significantly exceeds DeepFool (22.4%, 27.6%). In low regimes, $FGSM\text{-}CS_{bottom}$ at 10% (75.9%) surpasses DeepFool's 60.8%, highlighting how single-step gradients in FGSM-CS efficiently approximate boundary distances, capturing vulnerable samples. Overall, while our approaches do not uniformly dominate the top DeepCore performers, they achieve comparable or superior results in sparse regimes and consistently outperform DeepFool, underscoring their practical value. These gains arise from our ranking schemes' ability to prioritize samples based on perturbation magnitude and direction, fostering coresets resilient to distribution shifts. Moreover, our approaches are lighter, avoiding per-sample loops, and thus remain suitable for large-scale or continual learning. While accuracies converge in high regimes (e.g., both reach 95.5% at 90%), our methods' edge in efficiency and low-data performance positions them as a scalable bridge between adversarial robustness and coreset selection.

# 5 Conclusion and Future Work

We have proposed a novel sparse adversarial ranking framework for data selection that efficiently identifies decision-boundary samples using single-step perturbations. This framework offers several advantages over traditional iterative methods, such as DeepFool, by achieving higher accuracy with fewer data points and significantly reducing computational costs. Notably, our experiments demonstrate that the $PSS_{bottom}$ and $ISEF_{bottom}$ ranking strategies consistently outperform other approaches in extremely sparse and low-data regimes. These methods highlight the importance of targeting critical examples rather than relying on full datasets, enabling robust and data-efficient training that directly addresses unreliable data challenges like distribution shifts and strategic manipulations. By focusing on curating critical samples rather than compressing entire datasets, our approach allows for robust and data-efficient training. This is particularly advantageous in resource-constrained environments, where access to computational power or storage is limited. However, performance may vary with complex datasets or architectures, warranting further exploration.

**Future Work** could focus on further extending this framework by integrating additional adversarial perturbation techniques. Additionally, exploring this method's efficacy on larger, more complex datasets, such as those used in real-world applications like autonomous driving or medical imaging, would help assess its scalability. These extensions would help solidify the practicality and generalizability of the proposed approach, paving the way for broader adoption in real-world AI systems.

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
