# OpenReview forum: "Data-Efficient and Robust Coreset Selection via Sparse Adversarial Perturbations"
_NeurIPS.cc/2025/Workshop/Reliable_ML — NeurIPS 2025 - Reliable ML Workshop_

### Official Review · Reviewer_RHBy · 2025-09-08
**Questionable Generalizability and High Computational Cost**

**Rating:** 5
**Confidence:** 4

**Review:**

This paper integrates sparse adversarial perturbations into coreset selection. The authors use the Sparse Fast Gradient Sign Method to adversarially perturb training images and assess the sensitivity of a trained network to these perturbed samples. They propose three ranking methods to score samples based on their sensitivity to attacks, and the main contribution is combining these sensitivity metrics to guide coreset selection. The paper evaluates the performance of these ranking methods and compares them to various baselines in the literature. Overall, the idea of leveraging adversarial sensitivity for coreset selection is interesting and novel, but there are several concerns.

One concern is the heavy reliance on a single dataset (CIFAR-10) and the specific perturbation techniques used. The work would be more convincing if additional experiments or discussion were provided to demonstrate robustness across different datasets or alternative perturbation strategies.

Another issue is computational cost. The method requires training a model on the full dataset and computing predictions for all samples under multiple perturbation levels, which is non-trivial. Since coreset selection is intended to reduce computational burden, it would be helpful to discuss the tradeoff between this computational overhead and the benefits of coreset selection.

---

### Official Review · Reviewer_BRM5 · 2025-09-19
**Sparse-FGSM–guided coreset selection shows strong low-data gains but limited breadth and statistics**

**Rating:** 7
**Confidence:** 3

**Review:**

The authors propose an Adversarial Sensitivity Scoring framework with three ranking methods—ISEF, FGSM-CS, and PSS—that use single-step Sparse FGSM to prioritize boundary-proximal samples for coreset selection. The paper evaluates the approach on CIFAR-10 with ResNet-18 and reports large improvements over DeepFool in very low data regimes (e.g., up to ~15% absolute) while noting that “bottom” variants excel at tiny fractions and “top” variants overtake after ~20–30%.

Pros: The paper provides a clear method that leverages single-step sparse perturbations for efficiency, and it documents training and selection details (optimizer, schedule, fractions, and hyperparameters), which improves reproducibility. The experiments show competitive or superior accuracy to DeepFool and several DeepCore baselines in sparse/low regimes.

Cons: The paper evaluates only one dataset and one architecture, which limits generality. The manuscript emphasizes low overhead but does not quantify end-to-end wall-clock/GPU-hour costs, and it does not report multi-seed variance to establish statistical stability.

---

### Official Review · Reviewer_vNSQ · 2025-09-20
**Core-sets via adversarial robustness**

**Rating:** 4
**Confidence:** 2

**Review:**

## Summary

The paper proposes family of core-set selection techniques for training models on small amounts of data. The proposed family is derived from a combination of predictive uncertainty (standard for core-sets) and adversarial robustness. The idea is that by adding points to the coreset which if otherwise excluded would result in poor advesarial robustness, we can create a more sound core-set for general downstream tasks. The paper measures classification accuracy on CIFAR-10 by training ResNet-18 models on the coresets of each proposed and baseline method. They find that core-sets derived using techniques from adversarial robustness can compete with sota core-set techniques from DeepCore.

## Strengths

I do not work in core-sets / active learning, but the intuition of using adversarial robustness to strengthen core-set selection strategies is intuitive and natural. I like the idea.

The paper tests many different baselines, and one of the proposed methods seems to be competitive in each of the dataset fraction percentage regimes (e.g., using 0-10% of the data, 10-40% etc.).

##  Weaknesses

I could be missing something, but one major benefit of the proposed suite of methods is that the resulting models should have higher robust accuracy out of the box. This seems to be hinted at in the setup (line 219-224), but I do not seem to be able to find the results for this particular experiment.

Is there any way to test for statistical significance? By proposing so many new core-set methods, and testing on only a single dataset, one could imagine that some methods simply "get lucky" at certain dataset fractions for this particular training run. This is compounded for the extra small coresets with 0.1% of the dataset (~order of 50 samples). With such a small number of points in the coreset, I find it surprising that you would beat random, and indeed random does perform comparably here (Random is the top row of table 1). This weakens the claim that the proposed PSS_bottom method is doing something non-trivial.

In fact, I don't see any method uniformly outperforming random. The random baseline seems to be one of, if not the best method.

## Suggestions

1. Is there any way to combine the different methods from the family to create something which works uniformly across all regimes?

2. I am unconvinced by claim that the proposed methods improved over random selection. This could be biased based on my prior experience with active learning (similar to this coreset selection problem), where the random baseline is a very hard baseline to beat. Given that there is no statistical significance provided, I don't think that any claims about any of the methods being useful would stand up to scrutiny. I will, however, lower my confidence since this is outside of my area of expertise.

One small note, I saw some incorrect usage of \citep vs \cite for references. The APA has a style guide for these two types of references: https://www.unr.edu/writing-speaking-center/writing-speaking-resources/apa-7-in-text-citations#:~:text=Parenthetical%20citations%20include%20the%20author,publication%20(in%20parentheses)%20following.